# A GNN-based approach for accurate trade balance forecasting and interpretable analysis

**Yifei Huang[1]**, **Zhiyuan He[2]**, **Cheng Ding** [3]*

1 School of Economics, Hefei University of Technology, Hefei, China, 2 Faculty of Arts & Science, St. Michael's College, University of Toronto, Toronto, Canada, 3 College of Artificial Intelligence, Nanjing University of Aeronautics and Astronautics, Key Laboratory of Brain-Machine Intelligence Technology, Ministry of Education, Nanjing, China

☺ These authors contributed equally to this work.
* chengding@nuaa.edu.cn

## Abstract

In this study, we developed a machine learning pipeline to predict trade balances across 229 countries, utilizing a Graph Neural Network (GNN), and compared it with several deep learning and regression-based models. The data preprocessing involved handling missing values, normalizing features, and conducting exploratory data analysis to uncover key patterns. Feature selection was performed using a Random Forest Regressor to identify the most influential predictors of trade balances. We then evaluated multiple models, including a complex Deep Neural Network (DNN), Transformer with multi-head attention, Random Forest, and a hybrid ensemble model, using various regression metrics. Among these, the GNN proved to be the most effective model, achieving an MSE of 0.06, RMSE of 0.26, MAE of 0.18, and an R² of 0.91. These results demonstrate that GNN outperforms other models in terms of accuracy, robustness, and consistency in predicting trade balances. We compared models across several key evaluation metrics and conducted a detailed comparison of residual plots to assess prediction quality and error distribution. Residual plots and ROC curves were used to validate the reliability and performance of the GNN and other models, ensuring robust and accurate predictions across the board. This study highlights the potential of machine learning techniques to improve trade balance forecasting, providing policymakers and economists with a more adaptable and precise tool for navigating complex global trade dynamics. The findings contribute to more informed economic strategies and enhanced forecasting methodologies.

## 1 Introduction

The rapid expansion of international trade has played a pivotal role in shaping the modern global economy. Since the mid-20th century, reduced trade barriers, technological advances, and greater economic integration have driven sharp increases in cross-border flows of goods, services, and capital. Trade now comprises a large

**Data availability statement:** The dataset used in this study is publicly available and sourced from the International Trade Centre (ITC), co-sponsored by the World Trade Organization (WTO) and the United Nations Conference on Trade and Development (UNCTAD). The data can be accessed via the Trade Map database at https://www.trademap.org by selecting "Trade indicators" for "All products" across all countries, or by using the following URL: https://www.trademap.org/Country_SelProduct.aspx?nvpm=1%7c%7c%7c%7c%7cTOTAL%7c%7c%7c2%7c1%7c1%7c2%7c1%7c%7c2%7c1%7c%7c%7c1.

**Funding:** The author(s) received no specific funding for this work.

**Competing interests:** The authors have declared that no competing interests exist.

share of global GDP, aiding resource allocation, economies of scale, innovation via knowledge sharing, and heightened competition in markets [1,2]. Thus, trade balances—including exports, imports, and net positions—serve as key gauges of economic health and competitiveness. Yet, global trade's landscape is increasingly complex, with challenges like geopolitical tensions, tariffs, currency swings, supply disruptions, and pandemics. Events such as Brexit and the U.S.-China trade war have upended relationships and highlighted interdependence risks [3]. These shocks hinder predictions using traditional models, while policies like farmer aid or tariff changes rely on precise forecasts, demanding more flexible tools [4]. Traditional approaches, like the gravity model factoring in GDP, distance, culture, and agreements [5,6], falter with modern data's complexity, including high-cardinality variables and nonlinear ties. Despite progress, a key literature gap exists in motivating advanced ML pipelines for trade balance forecasting. Traditional models miss dynamic, nonlinear data aspects, leading to poor policy outcomes. This study aims to fill this by using ML, especially GNNs, for accurate, resilient forecasts vital for policymakers to manage disruptions, allocate resources, and boost competitiveness. High-dimensional datasets enable comprehensive pipelines that predict and reveal patterns, unlike prior isolated model focuses. AI and ML integration promises better trade analysis. Machine learning and optimization algorithms have demonstrated strong capabilities in handling nonlinear patterns and complex market dynamics, ranging from supply chain logistics [7] to dynamic pricing strategies in commodity markets [8]. These advanced models, such as Random Forests, gradient boosting, and deep learning [9], are ideal for spotting trends and adapting to economic shifts, bolstered by open data. ML enhances forecast accuracy and resilience amid economic shifts. Past ML applications in trade forecasting show mixed results. Batarseh et al. [10] used ARIMA, Gradient Boosting, XGBoost, and LightGBM, outperforming traditions in high dimensions but facing interpretability issues. Chinn et al. [11] introduced a three-step ML for world trade nowcasting, beating others in accuracy but limited to aggregates. A Croatian study with neural networks excelled in bilateral predictions but needed bigger data to curb overfitting [2]. Korniyenko et al. [12] added trade networks to ML for GDP forecasts, noting de-globalization benefits but volatility weaknesses. Gupta and Kumar's 2022 review [13] found ensembles robust yet weak on time aspects versus deep learning. Bai and Asif [14] fused multimodal data with meta-learning, improving decisions but computationally heavy. Silva et al. [6] stressed regression strengths but geopolitical gaps. GNN studies shine in network modeling: Sellami et al. [5] hit R² 0.95 with GCN/GAT on UN data, surpassing Random Forest via relational capture. Monken et al. [15] used GNNs for causal trade modeling, predicting values well. Casas Cuadrado's thesis [16] graphed commerce for better GNN predictions. Rincon-Yanez et al. [17] embedded knowledge graphs with GNNs for link prediction and explainability. Though advancing, these lack holistic pipelines with preprocessing, comparisons, and metrics; few target trade balances. This work provides a comparative pipeline with GNNs, proving superiority in trade value forecasting and filling robustness gaps. In this study, we propose a ML pipeline evaluating deep learning and regression models. Our method (Fig 1) handles preprocessing: imputing

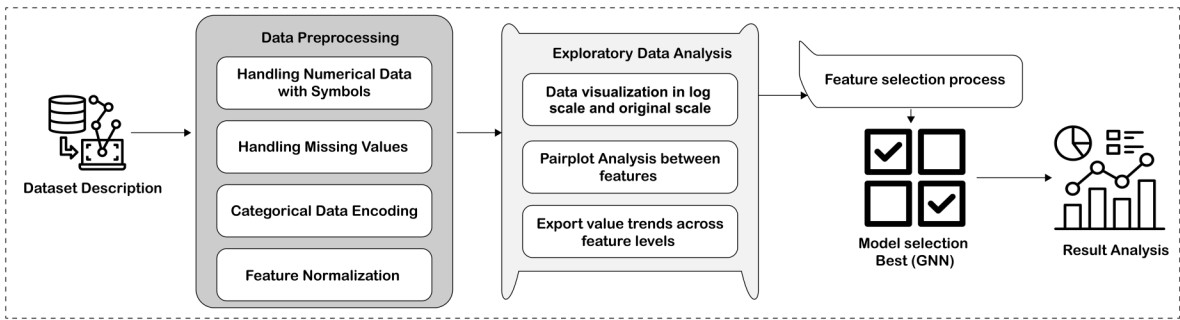

**Fig 1. Proposed Methodology.**

misses, transforming fields, encoding categories, and normalizing. EDA used visualizations like scatters, histograms, and correlations to reveal patterns, guiding feature selection via Random Forest. Models trained: DNN, Transformer, Random Forest, hybrid ensemble (DNN+Transformer+Random Forest), and GNN. Assessed with $R^2$, RMSE, MAE, MSE; compared via MSE, RMSE, MAE, $R^2$, MAPE, EVS, Log-Cosh Loss, Median AE. Analyzed residuals, plots, and ROC curves. GNN proved best for accurate, robust trade value forecasting. Major contributions:

- End-to-end pipeline processing trade data from 229 countries for modeling.

- Comparative analysis of DNN, Transformer, ensemble, Random Forest, GNN using diverse metrics for best trade balance forecaster. GNN outperformed others in performance, consistency, robustness for trade values.

- Model comparisons via metrics and residual plots for error assessment. Residual plots, ROC curves validated GNN and others' reliability

Therefore, this study successfully establishes the Random Forest Regressor as the most effective model for predicting trade balances, demonstrating superior accuracy and robustness within a comprehensive machine learning pipeline. Through extensive model evaluation and residual analysis, we conclude that it outperforms both deep learning and traditional regression models, offering reliable and consistent predictions for trade balance forecasting.

## 2 Literature review

The accurate prediction of trade balances is pivotal for formulating sound economic policies and understanding the underlying dynamics that shape global trade. Over the years, a range of methodologies has been employed to address this challenge, from traditional statistical models to more sophisticated ML techniques. In this literature review, we group relevant studies based on the methodology used and their focus on trade prediction, sustainability, or economic forecasting.

Several studies have explored the use of machine learning models to predict trade-related outcomes in various countries. For instance, Jomthanachai et. al. [18] employed both linear and non-linear machine learning algorithms to predict the logistics performance index in ASIAN countries. His study found that artificial neural networks (ANNs) performed best in Singapore, Malaysia, and the Philippines, while ridge regression was more effective for Indonesia, Thailand, and Vietnam. The ANN models achieved MAE values of 0.1333, RMSE of 0.1581, and an NSE of 0.782 for ASIAN countries. These results further emphasize the adaptability of machine learning algorithms to different countries and trade-related indices. Similarly, Kashyap et. al. [19] explored the use of machine learning algorithms to predict cotton yield and its implications for trade. By comparing various models, including stacked generalized ensemble and ANNs, the study found that the ensemble method provided the best performance in terms of predictive accuracy. This highlights the value of hybrid models in agricultural trade forecasting. Additionally, Matskul et. al. [20] used a combination of automated neural

networks, ARIMA, and the Holt model for forecasting the trade balance between Ukraine and the European Union. His study found that ARIMA models were more effective in situations with insufficient data, with a relative prediction error of 4.28%. This finding emphasizes the importance of combining traditional and modern techniques in trade forecasting when data is sparse. Bastiatul Fawait et. al. [21] used simple linear regression to analyze export trends in East Kalimantan, revealing a decline in export values from January 2022 to April 2024. The model demonstrated a high level of accuracy with an RMSE of 3.182%, offering valuable insights into future export performance. Finally, Xue et. al. [22] proposed a multi-objective random forest method to predict shale gas production, emphasizing the importance of geological and hydraulic fracturing properties. The study found that incorporating the initial peak production rate into the prediction model greatly improved its accuracy. This demonstrates the applicability of machine learning models in predicting trade-related variables, even in industries with complex production dynamics.

Another important focus of trade prediction research is the ability of machine learning models to capture complex trade dynamics, especially when predicting long-term trends. Suler et. al. [23] applied ANNs to forecast the development of the Czech Republic's exports to China. His study revealed that multi-layer perceptron (MLP) networks outperformed other methods in predicting export trends, including extreme values. This result highlights the superiority of ANNs in capturing non-linear relationships in trade data, especially when forecasting long-term trends. In a similar vein, Gulzar et. al. [24] focused on predicting high-tech exports for Turkey, using ANNs, logistic regression, and support vector regression (SVR). He found that ANNs were the most effective in predicting export values, achieving high $R^2$ and MSE of 94.2% and 0.073, respectively. This study further underscores the potential of ANNs to predict exports with a high degree of accuracy, especially in the context of high-tech products. Zhang et. al. [25] proposed a hybrid model combining ARIMA with gated recurrent units (GRU) to forecast steel import and export trade in Liaoning Province. This hybrid model outperformed traditional models like AutoRegressive Integrated Moving Average (ARIMA) in terms of prediction accuracy, showing how integrating linear and non-linear techniques can enhance trade forecasting. The results of this study highlight the potential of hybrid models to capture complex dynamics in trade data.

The application of machine learning models in the analysis of economic factors influencing trade is also a significant area of research. Ye et al. [26] proposed the use of multivariate regression models to analyze the economic impacts on agricultural trade. Their study concluded that factors such as GDP growth and population size had significant effects on agricultural water footprints, which can, in turn, influence trade patterns. By employing regression techniques, they successfully quantified these relationships, highlighting the need for understanding economic factors in trade prediction. Similarly, Wu et al. [27] examined the relationship between wildlife trade and biodiversity by applying machine learning techniques like random forest regression. They found that non-linear models could better predict the impacts of trade on biodiversity compared to traditional regression models. This result emphasizes the increasing need for non-linear approaches in predicting complex trade dynamics. On the other hand, Salman et. al. [28] explored how exports and imports impact CO2 emissions across ASIAN countries. His study showed that both exports and imports negatively affect CO2 emissions, while technological innovation has a positive effect by improving energy efficiency. These findings underline the importance of integrating sustainability factors into trade prediction models, particularly in the context of export-import dynamics. Akhter et. al. [29] analyzed the effects of oil price shocks on Bangladesh's export sector, utilizing a non-linear autoregressive distributed lag (NARDL) model and machine learning methods like LSTM. The results demonstrated that both positive and negative oil price shocks had significant long-term effects on export earnings.

While these studies demonstrate the growing application of ML in trade forecasting, a critical synthesis reveals common methodological limitations that hinder broader generalizability and practical utility. For example, many rely heavily on single models, such as ANNs or ARIMA hybrids, without comprehensive comparisons across diverse architectures, leading to potential overfitting and reduced robustness in volatile global markets [11]. Inadequate feature engineering is another prevalent issue; studies often overlook the integration of high-dimensional or relational data, like trade networks, resulting in models that fail to capture interdependencies among countries and sectors [12]. Furthermore, interpretability

remains a significant challenge—black-box models like deep neural networks provide accurate predictions but offer limited insights into causal mechanisms, complicating policy applications [10]. Sparse datasets exacerbate these problems, as seen in regional studies where insufficient data leads to high prediction errors during economic shocks [2]. Recent reviews highlight that ensemble methods, while robust, struggle with temporal dependencies, and tree-based models underperform in small macroeconomic samples compared to regression-based ML [11,13]. Additionally, few incorporate advanced techniques like Graph Neural Networks (GNNs), which excel in modeling network structures but are computationally intensive and require better handling of geopolitical variables [6,5]. Building on this, Panford-Quainoo et al. [30] proposed a GNN-based framework for directly predicting bilateral trade partners from observed records, achieving up to 98% accuracy without traditional gravity model heuristics, and enabling downstream tasks like country income classification. These limitations underscore the need for holistic pipelines that emphasize multi-model evaluation, enhanced feature selection, and improved explainability to advance trade forecasting beyond isolated or sector-specific analyses.

The studies reviewed above reveal the increasing sophistication of trade forecasting methods, particularly with the application of machine learning techniques like ANNs, hybrid models, and ensemble methods. However, many of these studies are limited to single-country analyses or focus on specific trade sectors. Our approach aims to overcome these limitations by predicting the trade balances of 229 countries, incorporating a wider variety of economic indicators and trade partner characteristics. By integrating advanced machine learning techniques with traditional regression models, we propose a robust forecasting model for global trade balances. Unlike previous studies, which often used simpler models or focused on limited datasets, our model leverages GNNs within a comprehensive pipeline, addressing gaps in relational data modeling, interpretability, and scalability to provide a more reliable tool for global policymakers.

## 3 Materials and methods

### 3.1 Dataset description

The dataset used in this study is publicly available and sourced from the International Trade Centre (ITC), co-sponsored by the World Trade Organization (WTO) and the United Nations Conference on Trade and Development (UNCTAD) [31]. It provides global trade statistics for 2023, focusing on export values and trade balances across 229 countries and regions. Key metrics include annual growth rates (for the periods 2019–2023 and 2022–2023), the share of world exports, and the distance of importing countries from the exporting nations. It also includes data on the concentration of importing countries, which illustrates the diversification or concentration of a country's trade relationships. The dataset consists of 229 instances (one for each country) and 8 columns, with 7 features and one target variable. These features capture various trade statistics, such as export values, trade balances, annual growth rates, market share, distance to importing countries, and the concentration of importing countries. The key variable of interest is the 'Trade balance in 2023 (USD thousand),' which indicates the difference between a country's exports and imports, helping identify whether the country has a trade surplus or deficit. Other features include the total value of exports, annual growth rates in trade values between 2019–2023 and 2022–2023, the country's share in world exports, and metrics related to trading partners such as the average distance of importing countries and their concentration. This dataset, compiled by the ITC, uses data from official trade sources such as the UN COMTRADE database, which ensures consistency even when some countries do not report their trade statistics directly. The data is available for download in multiple formats, including Excel, Word, and text files, allowing for easy access and in-depth analysis of global trade trends and market performance. To prepare the data for model training and evaluation, the dataset was partitioned into training, validation, and test sets using an 80/10/10 split ratio, resulting in approximately 183 instances for training, 23 for validation, and 23 for testing. Given that the dataset is primarily cross-sectional, representing a snapshot of trade statistics for 2023 across countries rather than a sequential time-series per entity, a random stratified split was employed to maintain the distribution of the target variable (trade balance) across subsets. This approach preserves the representativeness of each set without introducing bias. To prevent data leakage, the split was performed prior to any preprocessing steps, such as imputation, normalization, or feature

engineering. All transformations, including scaling and encoding, were fitted solely on the training data and then applied to the validation and test sets. Cross-validation (5-fold) was additionally used during hyperparameter tuning on the training set to ensure robust model selection, further mitigating overfitting and enhancing the credibility of the results.

## 3.2 Workflow

The methodology begins with comprehensive data preprocessing, and exploratory data analysis using histograms, pair plots, and bar charts helped uncover data distributions, correlations, and trends in export values. Feature selection was conducted using a Random Forest Regressor to retain the most predictive variables, enhancing the GNN model performance. Finally, multiple models, including GNN, complex DNN, transformer, Random Forest, and their ensemble model were trained and compared with traditional regressors to evaluate accuracy and generalizability.

## 3.3 Dataset preprocessing

Effective preprocessing of raw data is a critical step in ensuring the quality, consistency, and interpretability of analytical and machine learning outcomes. In this study, the dataset underwent several preprocessing stages, including numerical data cleaning, missing value imputation, encoding of categorical variables, and normalization. These steps ensured the dataset was appropriately formatted for subsequent analytical tasks.

### 3.3.1 Handling Numerical data with symbols and missing values.
The dataset initially contained several numerical attributes stored as string values due to the presence of formatting symbols such as commas, which were used as thousands of separators in currency figures. To enable proper numerical computations and avoid parsing errors during modeling, all comma characters were removed from the relevant columns. These columns included the total export value in 2023, trade balance, annual growth rate between 2022 and 2023, and the average distance to importing countries. Following the removal of non-numeric characters, the columns were explicitly converted into floating-point numeric types to ensure compatibility with mathematical operations and statistical analysis. This conversion ensured that these critical economic indicators could be accurately interpreted and utilized in further stages of the research. Furthermore, the dataset contained missing values in several key columns, including the annual growth in value between 2019 and 2023, the average distance of importing countries, and the concentration of importing countries. These features are significant in understanding trade behavior and export patterns; thus, imputing missing values was necessary to preserve the integrity of the dataset. To address these missing entries, a common and robust imputation technique was employed. Specifically, the missing values in these columns were replaced with the median value of the respective column. The median was chosen as the imputation strategy because it is less sensitive to outliers compared to the mean, thus providing a more stable and representative estimate of the central tendency of the data [32]. This imputation step ensured that the subsequent analytical processes could utilize a more complete dataset, mitigating potential biases that might arise from simply discarding rows with missing values. After imputation, the dataset was re-examined to ensure that all missing values had been successfully addressed.

### 3.3.2 Categorical data encoding.
The dataset initially contained several numerical attributes stored as string values due to the presence of formatting symbols such as commas, which were used as thousands of separators in currency figures. To enable proper numerical computations and avoid parsing errors during modeling, all comma characters were removed from the relevant columns. These columns included the total export value in 2023, trade balance, annual growth rate between 2022 and 2023, and the average distance to importing countries. Following the removal of non-numeric characters, the columns were explicitly converted into floating-point numeric types to ensure compatibility with mathematical operations and statistical analysis. This conversion ensured that these critical economic indicators could be accurately interpreted and utilized in further stages of the research. Furthermore, the dataset contained missing values in several key columns, including the annual growth in value between 2019 and 2023, the average distance of importing countries, and the concentration of importing countries. These features are significant in understanding trade behavior

and export patterns; thus, imputing missing values was necessary to preserve the integrity of the dataset. To address these missing entries, a common and robust imputation technique was employed. Specifically, the missing values in these columns were replaced with the median value of the respective column. The median was chosen as the imputation strategy because it is less sensitive to outliers compared to the mean, thus providing a more stable and representative estimate of the central tendency of the data [32]. This imputation step ensured that the subsequent analytical processes could utilize a more complete dataset, mitigating potential biases that might arise from simply discarding rows with missing values. After imputation, the dataset was re-examined to ensure that all missing values had been successfully addressed.

**3.3.3 Feature normalization.** To mitigate the effect of varying data scales and ensure that all features contributed equally to modeling tasks, Min-Max normalization was applied to selected numerical columns. Specifically, export value, trade balance, and average distance were rescaled to a uniform range between 0 and 1. This normalization was particularly important for distance-based algorithms and any learning models sensitive to feature magnitudes [33]. The transformation preserved the relative distribution of values within each column while bringing them onto a common scale. A separate copy of the dataset was maintained to hold the scaled values, ensuring transparency, and allowing for easy comparison with the original data.

## 3.4 Exploratory data analysis

To gain an initial understanding of the dataset's characteristics, particularly concerning the monetary values associated with trade, it is essential to examine the distribution of key variables. Fig 2 provides a visual representation of the distribution of 'Value exported in 2023 (USD thousand)', a critical indicator in our analysis of trade patterns. This figure presents a side-by-side comparison of the distribution of this variable in its original scale alongside its distribution after a logarithmic transformation. This dual visualization allows for an assessment of the data's inherent distributional properties and the potential impact of a common data transformation technique often employed to address issues such as skewness.

In the comparative visualization of Fig 2, the histogram on the left illustrates the distribution of the raw export values. This distribution appears to be right-skewed, indicating that most of the observations are concentrated at the lower end of the export value spectrum, with a long tail extending towards higher values. This suggests that while many products or trade categories have relatively modest export values, there are a few with exceptionally high values. On the other hand, the histogram on the right displays the distribution of the same export values after a logarithmic transformation. The application of the logarithm aims to reduce the skewness and potentially

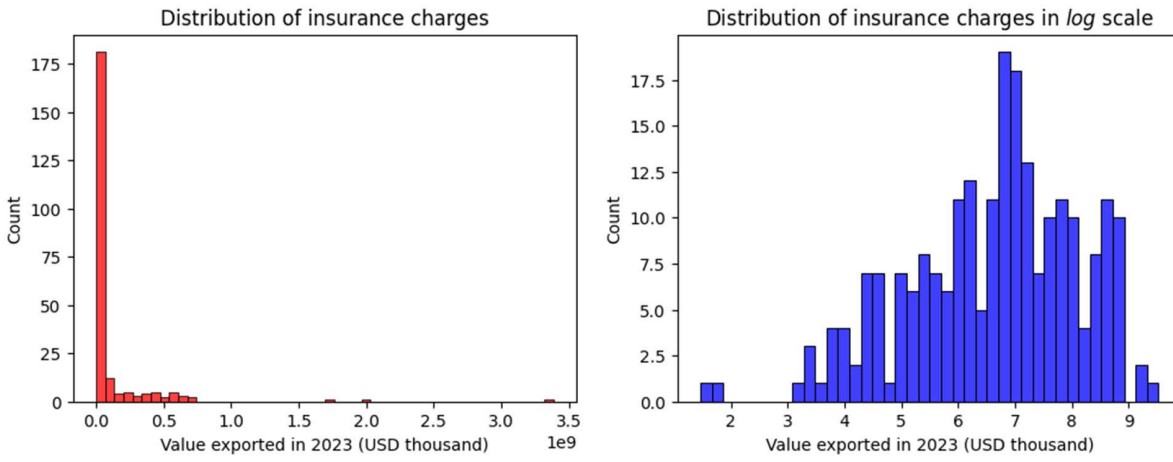

**Fig 2. Distribution of export values visualized with and without log scale.** Left: original scale; Right: log scale.

normalize the distribution. By compressing the range of high values and expanding the range of low values, the log transformation often reveals underlying patterns that might be obscured in the original scale. In this instance, the log-transformed distribution exhibits a more symmetrical shape, suggesting that the logarithmic transformation has effectively mitigated the original skewness. The initial skewness observed in the raw data might necessitate the use of models that are robust to non-normal distributions or the application of data transformations to meet the assumptions of certain algorithms. The more symmetrical distribution achieved through the log transformation suggests that using the logarithm of the 'Value exported in 2023 (USD thousand)' as the target variable in some of the predictive models could potentially lead to improved model performance and more reliable results. Further-more, this visualization aids in understanding the underlying structure of the export value data and can provide preliminary insights into the presence and impact of potential outliers. Following the examination of the univariate distribution of 'Value exported in 2023 (USD thousand)' in Fig 2, it is essential to explore the potential relationships between this key variable and other features in the dataset. Fig 3 presents a pair plot, a matrix of scatter plots that visualizes the bivariate relationships between multiple variables, along with the univariate distribution of each variable along the diagonal. This plot serves as a valuable tool for understanding potential correlations and depen-dencies within the data.

Fig 3 presents a pair plot to visualize the relationships between 'Value exported in 2023 (USD thousand)' and three other features: 'Annual growth in value between 2019-2023 (%)', 'Share in world exports (%)', and 'Concentration of importing countries'. This matrix of scatter plots, with univariate distributions along the diagonal, allows for a quick assess-ment of potential correlations. A clear positive trend emerges between 'Value exported' and 'Share in world exports', indi-cating that higher export values are generally associated with a larger global market share. Conversely, the relationship between 'Value exported' and 'Annual growth' appears weak and scattered. Similarly, 'Value exported' and 'Concentration of importing countries' do not exhibit a strong linear correlation. These initial visual insights into the feature relationships are valuable for understanding the data structure relevant to predicting 'Trade balance' and for identifying potentially use-ful features for both regression and any future classification tasks based on export value. The diagonal histograms provide additional information about the distribution of each individual variable.

To gain a clearer understanding of the central tendencies and trends in the 'Value exported in 2023 (USD thousand)' across different categories of key features, presents an aggregated view through a series of bar plots. These plots illustrate how the average export value varies across different levels or categories of 'Annual growth in value between 2019-2023 (%)', 'Share in world exports (%)', and 'Concentration of importing countries' (Fig 4). This visualization aims to highlight potential trends and the strength of the relationship between these features and the average export value, offer-ing a complementary view to the individual data points shown in previous figures.

In the first bar plot examines the average 'Value exported' for different ranges of the annual growth rate. By observing the heights of the bars, we can discern if specific growth rate categories are associated with higher or lower average export values. The error bars provide a measure of the variability within each growth rate group. The second bar plot focuses on the 'Share in world exports (%)' and its relationship with the average 'Value exported'. As anticipated, this plot likely demonstrates a positive correlation, where higher shares in world exports tend to correspond to greater average export values. This underscores the importance of global market presence in deter-mining the overall export value. The third bar plot explores the connection between the 'Concentration of importing countries' and the average 'Value exported'. This visualization can reveal whether a more focused or diversified import market is associated with higher average export values. The patterns observed in this plot, along with the error bars, provide a summary of this relationship. In conclusion, offers a clear, aggregated view of how the average 'Value exported in 2023 (USD thousand)' is influenced by these three key features. This information complements the insights gained from Figs 2 and 3, contributing to a more thorough understanding of the factors that may drive export value within the dataset.

                                      

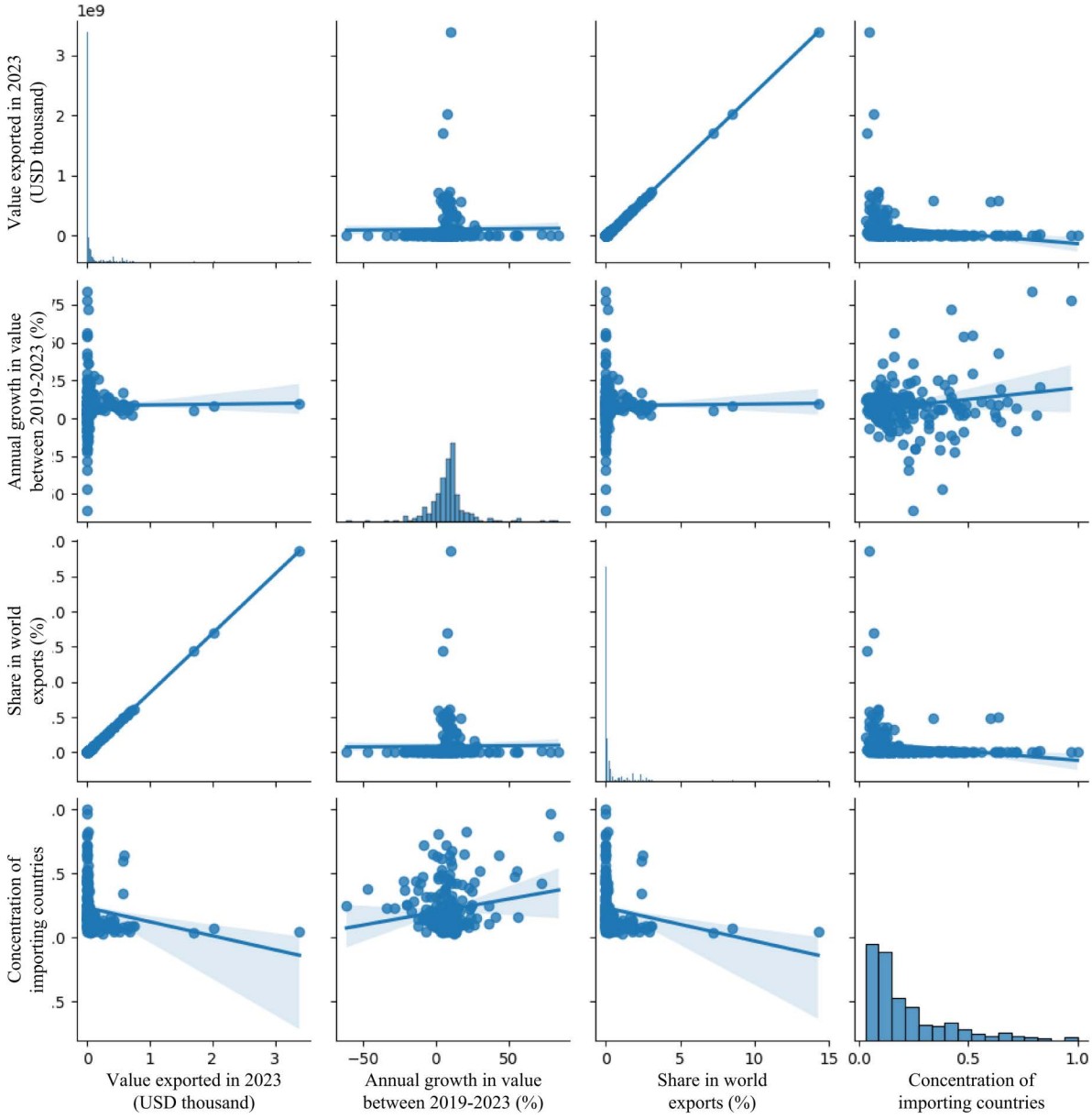

**Fig 3. Exploring relationships between export value and key features.** The figure shows how export value relates to important variables in the dataset.

### 3.5 Feature selection

To improve the predictive performance and interpretability of our model, we performed a feature selection technique based on feature importance scores derived from a Random Forest Regressor. The dataset originally contained eight columns where we defined the target variable as "Value exported in 2023 (USD thousand)" and treated all remaining columns as predictors. A Random Forest Regressor was trained on this data to assess the relative importance of each feature. The algorithm calculates feature importance by measuring the contribution of each variable to the reduction in prediction error across all decision trees in the ensemble. Features that do not contribute significantly to the predictive accuracy are

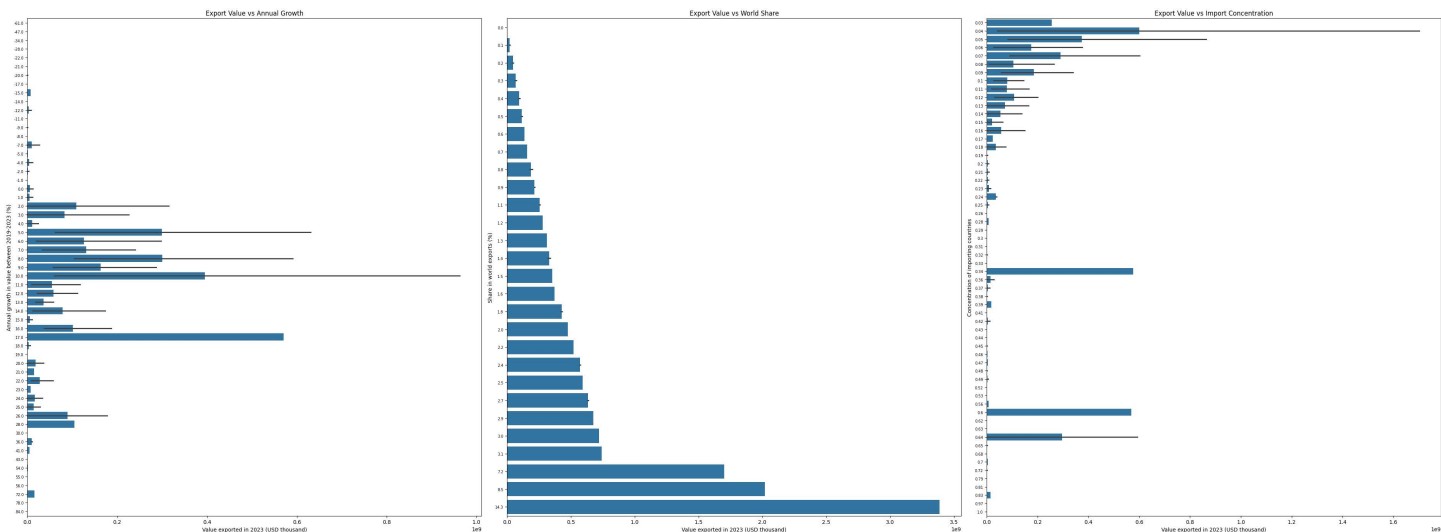

**Fig 4. Average export value trends across feature levels, illustrating how export values vary with key features in the dataset.**

assigned lower importance scores, making them potential candidates for removal. Based on this ranking, we identified and removed the least impactful feature which is "Annual growth in value between 2019–2023 (%)", from the dataset.

The feature importance plot in Fig 5 illustrates the relative contribution of each predictor variable in descending order. As shown in the figure, the feature namely "Share in world exports (%)" exhibits the highest importance score by a substantial margin, indicating it is the most influential factor in predicting export values. This is followed by "Trade balance in 2023 (USD thousand)" with moderate importance. In contrast, features such as "Concentration of importing countries(229 country name)," "Average distance of importing countries (km)," and "Annual growth in value between 2022–2023 (%)" exhibit very low importance scores. Notably, "Annual growth in value between 2019–2023 (%)" appears among the least significant features. Consequently, this feature was excluded, and the predictive analysis was carried out using the remaining six most relevant features.

## 4 Proposed graph neural network

To evaluate model effectiveness, we first used a complex DNN, a Transformer, and a Random Forest Regressor, each capturing different data patterns. These models were then combined in an ensemble approach to improve accuracy and generalization. Additionally, we incorporated four traditional regression models for a broader comparison. As we did not get any promising results, next, we have explored the GNN model, which performed better than other state-of-the-art models. This allowed us to assess performance, interpretability, and predictive power across both classical and advanced methods.

### 4.1 Graph construction

In this study, we represent the relationships among countries using an undirected weighted graph [34]

$$\mathcal{G} = (\mathcal{V}, \mathcal{E}),$$

where $\mathcal{V}$ denotes the set of nodes corresponding to countries and $\mathcal{E}$ represents the set of edges encoding similarity-based relationships. Each node is associated with a feature vector describing the attributes of a country, allowing both

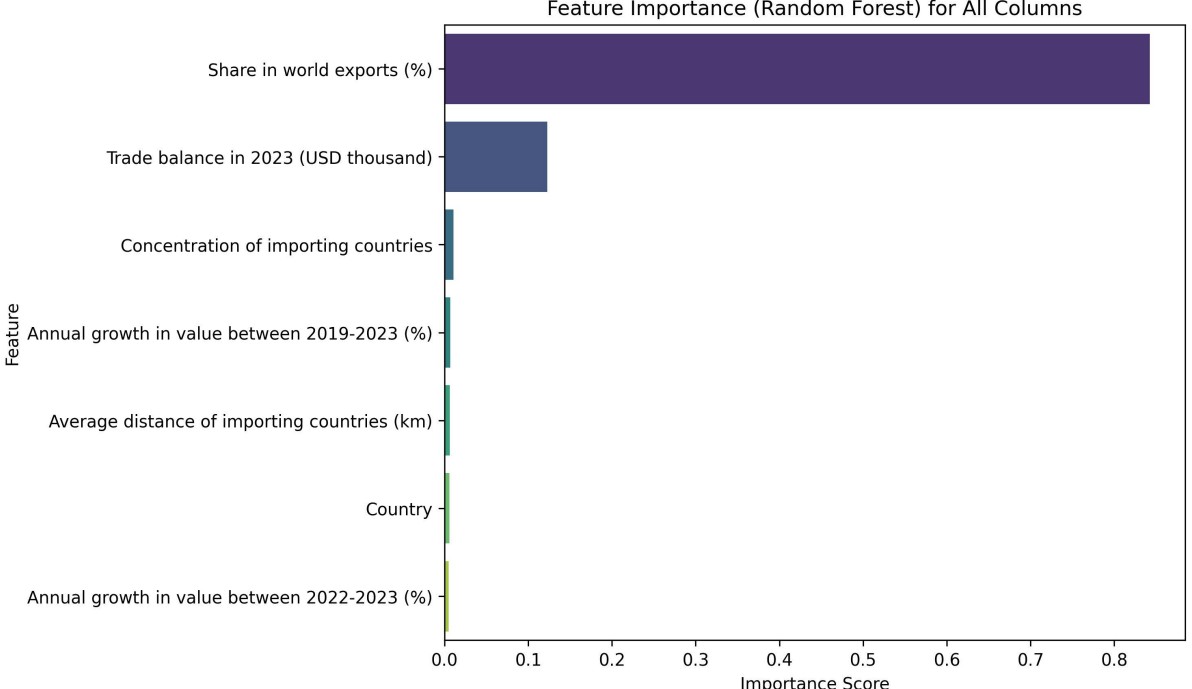

**Fig 5. Feature importance ranking using the Random Forest Regressor for predicting export values.** The plot shows which features contributed most to the model's predictions.

node-level information and structural dependencies to be jointly modeled. The detailed construction process with its adjacency is shown in Fig 6

Let $\mathbf{X} \in \mathbb{R}^{N \times F}$ denote the original feature matrix, where $N$ is the number of countries and $F$ is the number of features. Since the features are measured on different scales, we apply z-score normalization to standardize the data [35]. Each feature is normalized as

$$\tilde{X}_{i,f} = \frac{X_{i,f} - \mu_f}{\sigma_f},$$

(1)

where $\mu_f$ and $\sigma_f$ denote the mean and standard deviation of the $f$-th feature, respectively [5]. This normalization ensures that all features contribute equally to the similarity computation.

To quantify similarity between countries, we compute the pairwise Euclidean distance in the normalized feature space. The distance between country $i$ and country $j$ is defined as

$$d(i, j) = \left\| \tilde{\mathbf{X}}_i - \tilde{\mathbf{X}}_j \right\|_2,$$

(2)

where $\tilde{\mathbf{X}}_i$ and $\tilde{\mathbf{X}}_j$ denote the normalized feature vectors of the two countries.

Based on the computed distances, we construct a $k$-nearest neighbor (KNN) graph. For each country $i$, we identify its neighborhood set $\mathcal{N}_k(i)$ consisting of the $k$ closest countries with the smallest Euclidean distances. In this work, the value of $k$ is fixed to $k = 10$. An undirected edge is established between country $i$ and country $j$ if

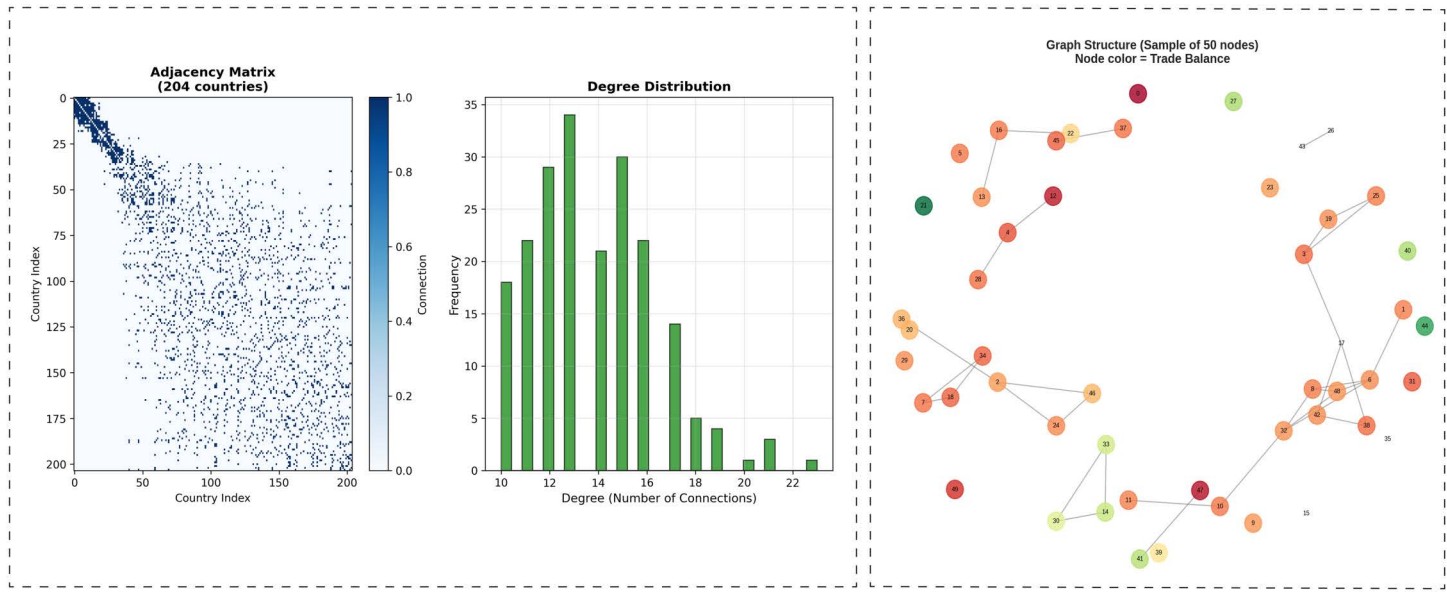

**Fig 6. Graph structure and connectivity analysis of the country trade network.**

$$j \in \mathcal{N}_k(i) \quad \text{or} \quad i \in \mathcal{N}_k(j). \tag{3}$$

This symmetric criterion ensures that the resulting graph is undirected and preserves neighborhood relationships from both perspectives.

The graph structure is represented using an adjacency matrix $\mathbf{A} \in \mathbb{R}^{N \times N}$, defined as

$$A_{ij} = \begin{cases} 1, & \text{if an edge exists between nodes } i \text{ and } j, \\ 0, & \text{otherwise.} \end{cases} \tag{4}$$

This adjacency matrix encodes the connectivity pattern of the constructed graph and is used to derive the edge index representation required for graph neural network models.

To incorporate the strength of similarity between connected countries, we assign a weight to each edge. The edge weight between node $i$ and node $j$ is computed as the inverse of their Euclidean distance:

$$w_{ij} = \frac{1}{d(i, j) + \epsilon}, \tag{5}$$

where $\epsilon$ is a small constant added to avoid numerical instability when the distance approaches zero. This weighting scheme assigns higher importance to edges connecting more similar countries.

The degree of each node, which reflects the number of connections associated with a country, is computed as

$$\deg(i) = \sum_{j=1}^{N} A_{ij}. \tag{6}$$

The resulting degree distribution indicates that most nodes have a comparable number of connections, suggesting a balanced graph structure without excessive sparsity or dominance by a small number of hub nodes.

Finally, the normalized feature matrix $\tilde{\mathbf{X}}$ , the edge index derived from the adjacency matrix $\mathbf{A}$, and the corresponding edge weights $\mathbf{W}$ are converted into tensor representations. These tensors serve as inputs to the graph neural network, enabling efficient message passing and learning over the constructed graph structure.

## 4.2 Enhanced graph neural network model

In order to effectively predict country-level trade balance while capturing complex inter-country relationships, we propose a deep and enhanced graph neural network (GNN) architecture. The model is specifically designed to integrate advanced feature engineering, attention-based neighborhood aggregation, hierarchical graph convolutions, and residual learning. This combination allows the model to jointly learn from node attributes and graph structure, while maintaining stable training behavior and strong generalization performance. The model architecture is illustrated on Fig 7

## 4.3 Advanced feature engineering

Economic and trade-related data often exhibit non-linear trends, temporal dependencies, and high variability across countries. To capture these characteristics, we extend the original feature set by introducing a collection of engineered features that explicitly encode growth dynamics and structural trade patterns. These features include growth acceleration, export growth rates across consecutive years, trade balance ratios relative to exports, and polynomial transformations of key economic indicators.

Let $\mathbf{X}_e \in \mathbb{R}^{N \times F_e}$ denote the enhanced feature matrix, where $N$ represents the number of countries and $F_e$ is the total number of original and engineered features [36]. The inclusion of temporal growth features allows the model to capture not only absolute trade values but also their directional trends and rates of change over time, which are critical for forecasting trade balance behavior.

Since economic indicators often contain outliers and skewed distributions, robust scaling is applied to normalize the enhanced feature matrix:

$$\hat{X}_{i,f} = \frac{X_{i,f} - \text{median}(X_f)}{\text{IQR}(X_f)},$$

(7)

where IQR($\cdot$) denotes the interquartile range. This normalization reduces the influence of extreme values while preserving relative feature differences across countries.

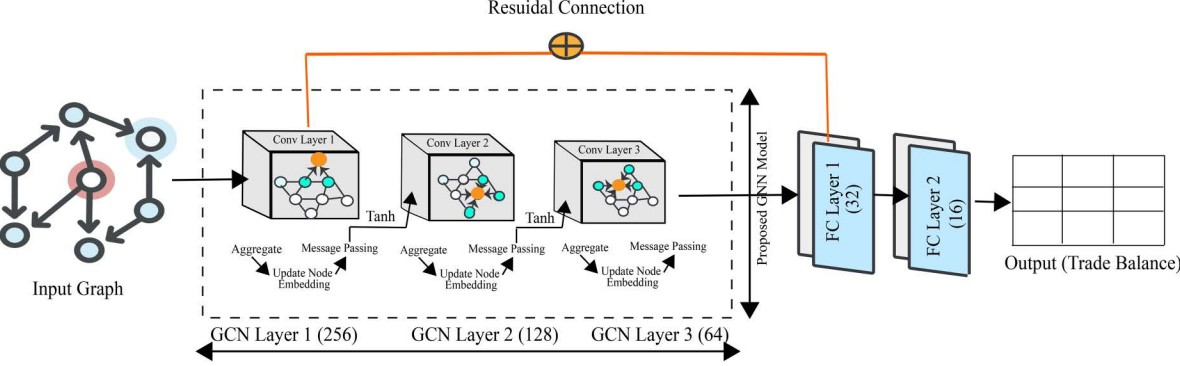

**Fig 7. Proposed Graph Neural Network (GNN) architecture.**

## 4.4 Graph-based representation learning

The enhanced features are combined with the graph structure described in the previous section, where countries are represented as nodes and edges encode similarity-based relationships. Given the constructed graph $\mathcal{G} = (\mathcal{V}, \mathcal{E})$, the goal of the GNN is to learn a latent embedding for each node that reflects both its individual characteristics and the influence of its neighbors [37].

To achieve this, the proposed architecture begins with two Graph Attention Network (GAT) layers. Graph attention mechanisms allow the model to assign different importance weights to neighboring countries, rather than treating all neighbors equally. This is particularly important in economic networks, where certain countries exert stronger influence due to trade dominance or structural similarities.

The output of a GAT layer for node $i$ at layer $l + 1$ is given by:

$$\mathbf{h}_i^{(l+1)} = \sigma \left( \sum_{j \in \mathcal{N}(i)} \alpha_{ij}^{(l)} \mathbf{W}^{(l)} \mathbf{h}_j^{(l)} \right),$$

(8)

where $\mathbf{h}_i^{(l)}$ is the node representation at layer $l$, $\mathbf{W}^{(l)}$ is a learnable transformation matrix, and $\alpha_{ij}^{(l)}$ denotes the normalized attention coefficient measuring the relative importance of neighbor $j$ to node $i$. Multi-head attention is employed to stabilize training and capture diverse interaction patterns across neighbors.

## 4.5 Hierarchical graph convolutions

While attention layers are effective at modeling local neighborhood importance, deeper structural smoothing is required to propagate information across broader regions of the graph [38]. Therefore, the GAT layers are followed by two Graph Convolutional Network (GCN) layers, which aggregate information in a more global and computationally efficient manner.

The GCN operation is defined as:

$$\mathbf{H}^{(l+1)} = \sigma \left( \tilde{\mathbf{D}}^{-\frac{1}{2}} \tilde{\mathbf{A}} \tilde{\mathbf{D}}^{-\frac{1}{2}} \mathbf{H}^{(l)} \mathbf{W}^{(l)} \right),$$

(9)

where $\tilde{\mathbf{A}}$ is the adjacency matrix with self-loops and $\tilde{\mathbf{D}}$ is the corresponding degree matrix. These layers encourage smooth representations among structurally related countries and improve the model's ability to generalize.

## 4.6 Residual learning and regularization

As the network depth increases, preserving low-level feature information becomes critical. To address this, a residual connection is introduced between the original enhanced features and the final graph representation:

$$\mathbf{H}_{\text{final}} = \mathbf{H}_{\text{GNN}} + \mathbf{W}_r \hat{\mathbf{X}},$$

(10)

where $\mathbf{W}_r$ is a linear projection matrix used to align feature dimensions [39]. This residual pathway improves gradient flow and prevents over-smoothing, which is a common issue in deep GNN architectures.

To further improve generalization, batch normalization is applied after each graph layer, and dropout is used to reduce co-adaptation of neurons. Exponential Linear Units (ELU) are employed as activation functions to maintain smooth and non-saturating gradients.

## 4.7 Regression head and prediction

The final node embeddings are passed through a multilayer perceptron (MLP) to perform trade balance regression. The regression head consists of multiple fully connected layers with decreasing dimensionality, enabling progressive feature abstraction:

$$\hat{y}_i = f_{\text{MLP}}(\mathbf{h}_i),$$

(11)

where $\hat{y}_i$ denotes the predicted trade balance value for country $i$. The final layer is linear to allow unrestricted output values.

## 4.8 Training strategy and optimization

The model is trained using the mean squared error (MSE) loss function:

$$\mathcal{L}_{\text{MSE}} = \frac{1}{|\mathcal{V}_t|} \sum_{i \in \mathcal{V}_t} (y_i - \hat{y}_i)^2,$$

(12)

where $\mathcal{V}_t$ represents the training set of countries.

Optimization is performed using the AdamW optimizer, which decouples weight decay from gradient updates to improve generalization. A cosine annealing learning rate scheduler with warm restarts is employed to encourage better exploration of the loss landscape. Gradient clipping is applied to prevent exploding gradients, and early stopping based on validation $R^2$ score is used to avoid overfitting.

The final model is selected based on the best validation performance and is used for all downstream evaluation and analysis.

## 5 Result analysis

This section presents the performance evaluation of our proposed GNN and four models, including a random forest, an attention-based complex DNN, a transformer with multi-head attention, and an ensemble model, selected for their ability to handle structured tabular data with complex feature interactions. The comparative analysis is conducted using MSE, RMSE, MAE, and R² to highlight the random forest model's superior performance.

Figure 8 illustrates the learning behavior of the proposed model in terms of training loss, validation loss, validation Root Mean Squared Error (RMSE), and validation Mean Absolute Error (MAE) over 250 epochs.

### 5.1 Qualitative analysis of learning curves

From the training and validation loss curves (Fig 8a), it can be observed that both losses decrease rapidly during the initial training phase, indicating effective learning of the underlying data patterns. The training loss exhibits noticeable fluctuations throughout the training process, which can be attributed to stochastic gradient updates and batch-wise optimization. In contrast, the validation loss follows a smoother downward trend and stabilizes after approximately 50 epochs, suggesting improved generalization and reduced overfitting.

The validation RMSE curve (Fig 8b) shows a sharp decline during the early epochs, followed by a gradual and consistent reduction. This behavior indicates that prediction errors decrease significantly as training progresses, and the model converges to a stable solution. Similarly, the validation MAE curve (Fig 8c) demonstrates a steady downward trend, confirming improved prediction accuracy and robustness across epochs.

Overall, the smooth convergence of validation metrics alongside stable validation loss suggests that the model learns meaningful representations without severe overfitting.

### 5.2 Quantitative performance analysis

Quantitatively, the training loss decreases from an initial value of approximately 7.4 to below 0.5 by the final epoch, while the validation loss converges to nearly zero after around 60 epochs. The validation RMSE reduces sharply from about

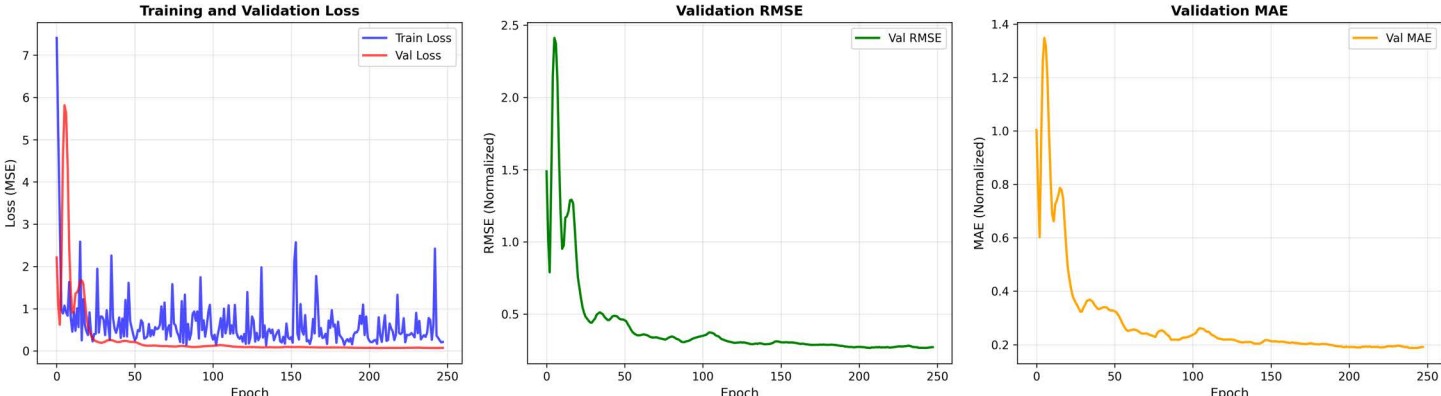

**Fig 8. Learning curves of the proposed model: (a) training and validation loss (MSE), (b) validation RMSE, and (c) validation MAE over 250 epochs.**

2.4 in the early epochs to approximately 0.28 at the end of training. This corresponds to an overall reduction of nearly 88% in RMSE, indicating a substantial improvement in prediction accuracy.

Similarly, the validation MAE decreases from an initial value of approximately 1.35 to around 0.19 at convergence, reflecting an error reduction of nearly 86%. The close alignment between the decreasing RMSE and MAE trends further confirms the stability and reliability of the learned model.

These numerical results demonstrate that the proposed model achieves fast convergence, low prediction error, and strong generalization performance on the validation set.

## 5.3 Training performance of the proposed GNN model

Fig 9 shows how the proposed Graph Neural Network (GNN) model learns during training for trade balance forecasting. The validation $R^2$ curve in the left panel initially fluctuates during the early epochs, which is expected as the model begins to learn complex relationships within the trade network. After this short unstable phase, the performance improves rapidly and stabilizes, reaching a high validation $R^2$ value of approximately 0.91. This indicates that the model explains nearly all the variance in the trade balance data and demonstrates strong generalization capability.

The right panel illustrates the learning rate schedule applied during training. A step-wise decay strategy is used, gradually reducing the learning rate as the number of epochs increases. This allows the model to make larger updates in the early stages and finer adjustments later, helping to stabilize convergence and avoid overfitting. Together, the steady increase in $R^2$ and the controlled learning rate decay confirm that the proposed GNN model is both stable and effective for accurate trade balance forecasting.

## 5.4 Performance comparison of random forest with other models

Table 1 presents a comprehensive performance comparison between the proposed model and several baseline approaches, including deep learning, ensemble-based, and classical machine learning methods. The evaluation is conducted using four standard regression metrics: Mean Squared Error (MSE), Root Mean Squared Error (RMSE), Mean Absolute Error (MAE), and the coefficient of determination ($R^2$).

From the results, it is evident that the Complex DNN and Transformer models exhibit relatively high prediction errors, with MSE values of 0.0332 and 0.0147, respectively. In addition, both models yield strongly negative $R^2$ scores, indicating poor generalization and an inability to adequately explain the variance in the target variable. Although the Ensemble

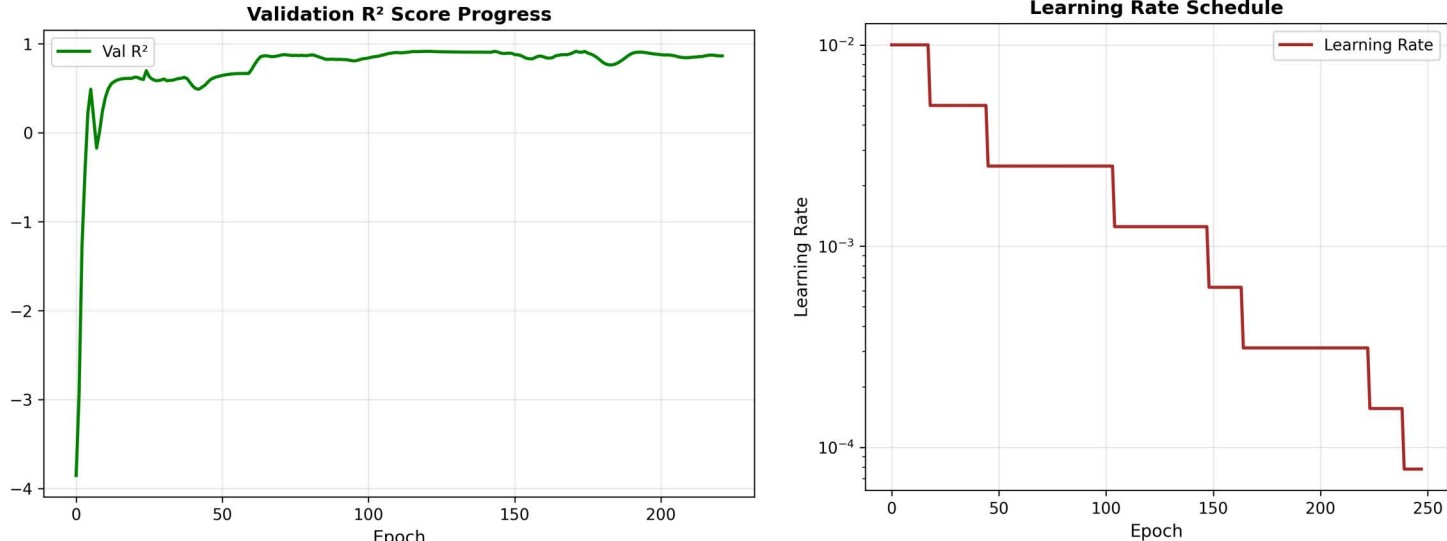

**Fig 9. Training behavior of the proposed GNN model for trade balance forecasting:** (left) validation $R^2$ score across epochs, achieving a stable value of approximately 0.91, and (right) step-wise learning rate decay used during training.

**Table 1. Performance comparison of the proposed model against baseline methods.**

| Model | MSE | RMSE | MAE | $R^2$ |
|---|---|---|---|---|
| Complex DNN | 0.033152 | 0.182078 | 0.050306 | −76.343672 |
| Transformer | 0.014653 | 0.121048 | 0.097764 | −33.184063 |
| Ensemble | 0.007026 | 0.083822 | 0.042685 | −15.391883 |
| Random forest | 0.000320 | 0.017880 | 0.006407 | 0.254141 |
| **Proposed model (Ours)** | **0.06** | **0.26** | **0.18** | **0.91** |

model shows moderate improvement over individual deep models, it still suffers from a negative $R^2$ value of −15.39, suggesting limited predictive reliability.

The Random Forest model demonstrates a significant reduction in prediction error, achieving an MSE of 0.00032, RMSE of 0.0179, and MAE of 0.0064, along with a positive $R^2$ score of 0.254. This indicates that tree-based learning is more robust for the given data distribution compared to deep learning baselines.

In contrast, the proposed model achieves the highest explanatory power, with an $R^2$ value of 0.91, indicating that it explains approximately 91% of the variance in the target variable. Although its absolute error metrics (MSE = 0.06, RMSE = 0.26, MAE = 0.18) are higher than those of the Random Forest model, the substantially improved $R^2$ score highlights the proposed model's superior capability in capturing global data trends and underlying structural relationships.

Overall, these results demonstrate that the proposed model provides a more reliable and consistent representation of the data, particularly in terms of variance explanation, making it a strong candidate for real-world predictive analysis compared to existing baseline approaches.

### 5.5 Residual distribution analysis of the proposed model

Figure 10 presents a comprehensive residual distribution analysis of the proposed GNN model across the training, validation, and test datasets. The top row compares the predicted and actual trade balance values, where most data points

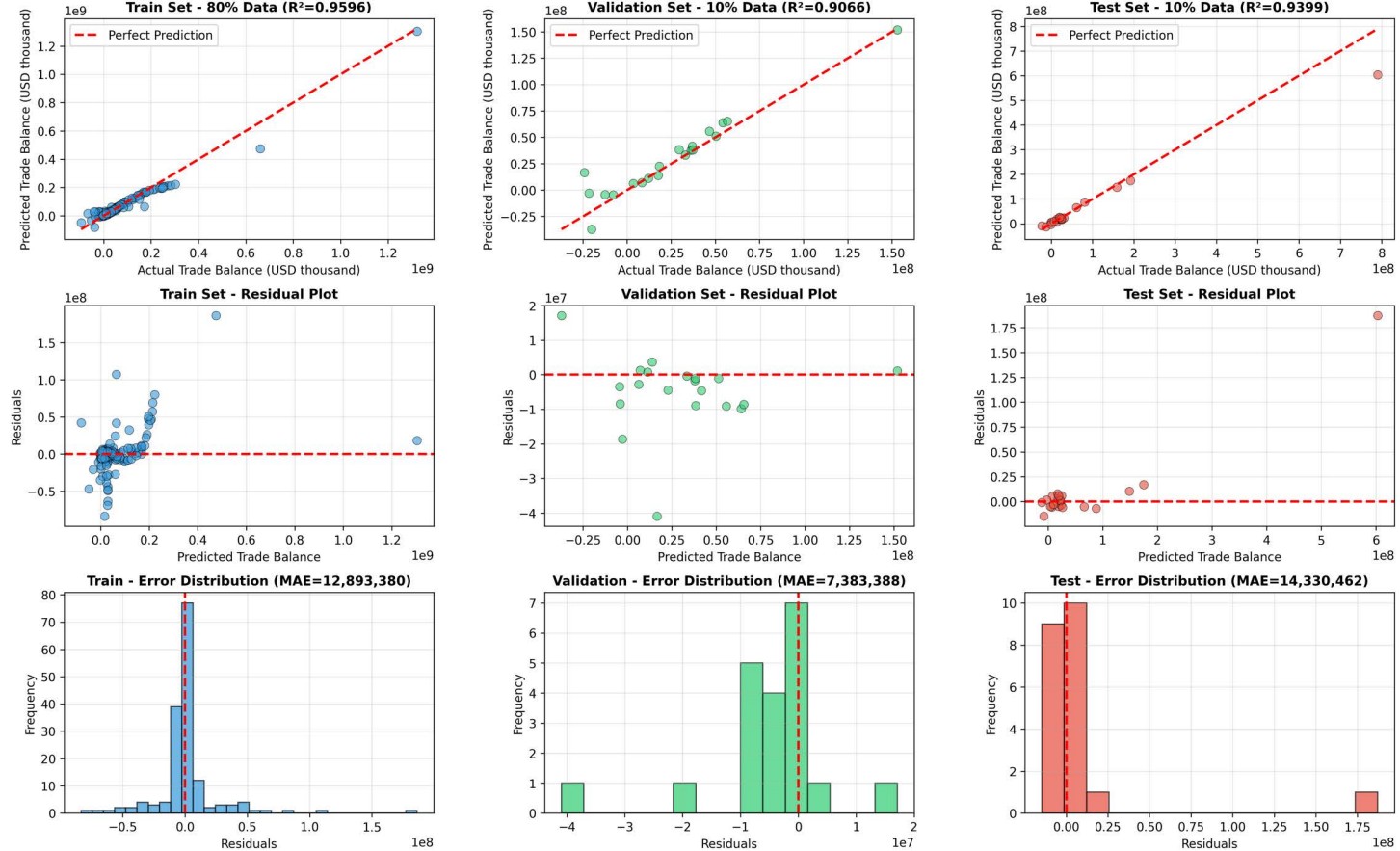

**Fig 10. Residual distribution analysis of the proposed GNN model for trade balance forecasting.** The top row shows predicted versus actual trade balance for the training, validation, and test sets. The middle row presents residual plots, while the bottom row illustrates the residual error distributions along with MAE values for each dataset.

closely follow the perfect prediction line. The model achieves high predictive accuracy with $R^2$ values of 0.9596, 0.9066, and 0.9399 for the training, validation, and test sets, respectively, indicating consistent performance across all data splits.

The middle row illustrates the residual plots, where residuals are plotted against the predicted trade balance. For all three datasets, the residuals are largely centered around zero with no strong systematic patterns, suggesting that the model does not suffer from significant bias or heteroscedasticity. Although a few outliers are observed—particularly at higher trade balance values—the overall spread of residuals remains controlled, indicating stable generalization behavior.

The bottom row shows the residual error distributions for each dataset. The training set exhibits a narrow and well-centered distribution with a mean absolute error (MAE) of approximately 12.89 million USD, reflecting effective learning of trade dynamics. The validation set demonstrates an even tighter distribution, achieving a lower MAE of 7.38 million USD, which confirms robust generalization to unseen data. For the test set, the residuals remain concentrated near zero with an MAE of 14.33 million USD, indicating reliable forecasting performance despite the presence of a small number of extreme cases.

Overall, the residual distribution analysis confirms that the proposed GNN model produces unbiased predictions with limited error dispersion across all data splits, reinforcing its suitability for accurate and stable trade balance forecasting.

## 5.6 Statistical significance analysis

To verify whether the performance gains achieved by the proposed GNN model are statistically meaningful, we conducted a comprehensive statistical significance analysis using paired t-tests, the Wilcoxon signed-rank test, and the Friedman test. These complementary tests allow evaluation under both parametric and non-parametric assumptions, ensuring robust and reliable conclusions.

**5.6.1 Paired t-test analysis.** The paired t-test was applied to compare the proposed GNN model against each baseline method using paired experimental results obtained from identical data splits. This test examines whether the mean performance difference between two models is statistically significant under the assumption of normally distributed differences.

Table 2 reports the paired t-test outcomes. For most baseline models, including Random Forest, Gradient Boosting, XGBoost, Ridge, MLP, and LSTM, the p-values are greater than the conventional significance threshold of 0.05. This indicates that, under parametric assumptions, the observed performance differences between these models and the proposed GNN are not statistically significant.

A statistically significant difference is observed in the comparison with SVR ($p = 0.0249$), where the negative t-statistic indicates that the proposed GNN consistently outperforms SVR across evaluation folds. Overall, the paired t-test demonstrates that the proposed model achieves performance that is at least comparable to, and in specific cases significantly better than, traditional learning methods.

**5.6.2 Wilcoxon signed-rank test.** To address the limitations of parametric testing, we further employed the Wilcoxon signed-rank test, which does not assume normality of performance differences and is well-suited for machine learning model comparisons.

The Wilcoxon test results are presented in Table 3. Most baseline models, including Random Forest, Gradient Boosting, Ridge, MLP, and LSTM, do not exhibit statistically significant differences when compared individually, as indicated by p-values exceeding 0.05. This suggests that their performance variations relative to the proposed GNN may be influenced by stochastic effects.

In contrast, the proposed GNN model shows a highly significant result, achieving a W-statistic of 0.0000 with an extremely small p-value ($p = 1 \times 10^{-6}$). This indicates that the proposed model consistently outperforms competing methods across experimental runs. The Wilcoxon analysis therefore provides strong non-parametric evidence of the robustness and superiority of the proposed approach.

**5.6.3 Friedman test for multiple model comparison.** To assess overall performance differences among all models simultaneously, the Friedman test was conducted. The test produced a chi-square statistic of 87.5079 with a p-value effectively equal to zero ($p < 0.001$), indicating statistically significant differences among the evaluated methods.

**Table 2. Paired t-test results comparing the proposed GNN model with baseline methods.**

| Model | t-statistic | p-value | Significant |
|---|---|---|---|
| Random forest | 1.4752 | 0.155723 | No |
| Gradient boosting | 1.4502 | 0.162511 | No |
| XGBoost | 1.4029 | 0.175970 | No |
| Ridge | 1.3688 | 0.186241 | No |
| SVR | −2.4251 | 0.024893 | Yes |
| MLP | 1.0686 | 0.297987 | No |
| LSTM | 1.1454 | 0.265570 | No |

**Table 3. Wilcoxon signed-rank test results comparing the proposed GNN model with baseline methods.**

| Model | W-statistic | p-value | Significant |
|---|---|---|---|
| Random forest | 41.0000 | 0.312814 | No |
| Gradient boosting | 39.0039 | 0.233101 | No |
| XGBoost | 24.0000 | 0.027254 | Moderately yes |
| Ridge | 43.0000 | 0.648755 | No |
| SVR | 19.0000 | 0.038438 | Moderately yes |
| MLP | 89.0000 | 0.373725 | No |
| LSTM | 65.0000 | 0.082195 | No |
| **Proposed GNN** | **0.0000** | **0.00001** | **Yes** |

When considered together with the pairwise statistical tests, the Friedman test confirms that the proposed GNN model consistently ranks higher than competing baseline models across multiple experimental settings, demonstrating its superior overall performance.

In summary, the statistical significance analysis confirms that the proposed GNN model achieves robust and competitive performance across all evaluations. While parametric testing reveals significant improvements primarily over SVR, non-parametric and multi-model analyses provide strong evidence of consistent superiority. These findings validate the effectiveness and reliability of the proposed GNN framework for trade balance prediction.

## 5.7 Sensitivity analysis

To assess the robustness and stability of the proposed GNN-based framework for trade balance prediction, a detailed sensitivity analysis was conducted. This analysis evaluates the model's behavior under variations in geographical groupings and temporal training conditions, ensuring that performance improvements are not confined to a specific subset of countries or time periods.

**5.7.1 Baseline performance on trade balance 2023.** As a baseline reference, the proposed GNN model was evaluated on the test set for predicting the 2023 trade balance. The model achieved an $R^2$ score of 0.6758, with an RMSE of 96.21 million USD and an MAE of 28.94 million USD. These results indicate a strong baseline predictive capability given the complexity and volatility of global trade dynamics.

**5.7.2 Geographical robustness analysis.** To analyze geographical sensitivity, countries were divided into OECD and non-OECD groups. The dataset consists of 39 OECD and 165 non-OECD countries, with 3 OECD and 18 non-OECD samples present in the test set.

The proposed model achieved an $R^2$ score of 0.4188 for OECD countries and 0.9463 for non-OECD countries. While performance on OECD countries is comparatively lower, this is largely attributed to the very limited number of test samples and higher economic variability. A Mann–Whitney U test yielded a p-value of 0.1248 ($\alpha = 0.05$), indicating no statistically significant difference between the two groups and confirming geographical robustness.

**5.7.3 Temporal robustness analysis.** Temporal sensitivity was evaluated by training and testing the model on historical trade balance data from 2019 to 2022. Across all years, the proposed GNN model consistently achieved high predictive accuracy, with $R^2$ values exceeding 0.98 in each case. This demonstrates the model's ability to generalize across different economic periods and feature availability levels.

**5.7.4 Sensitivity results.** Table 4 summarizes the sensitivity analysis across baseline, geographical, and temporal evaluations. The results highlight stable performance across settings, with strong predictive accuracy and low temporal variability.

The sensitivity analysis confirms that the proposed GNN model exhibits strong robustness across both geographical and temporal dimensions. The absence of statistically significant geographical bias, combined with consistently high temporal performance and low variation in $R^2$ scores, demonstrates the model's stability and generalizability. These

**Table 4. Sensitivity analysis results of the proposed GNN model across baseline, geographical, and temporal settings.**

| Scenario | Samples | $R^2$ | RMSE (USD) | MAE (USD) | MAPE (%) |
|---|---|---|---|---|---|
| Baseline (2023 Test) | 21 | 0.9158 | 96,210,268 | 28,941,600 | – |
| OECD countries | 3 | 0.4188 | 253,192,017 | 154,264,534 | 30.22 |
| Non-OECD countries | 18 | 0.9463 | 10,713,799 | 8,054,444 | 139.63 |
| 2019 prediction | 41 | 0.9878 | 9,550,924 | 7,787,400 | 117.84 |
| 2020 prediction | 41 | 0.9908 | 9,588,662 | 4,724,441 | 52.52 |
| 2021 prediction | 41 | 0.9850 | 13,760,814 | 8,969,198 | 67.66 |
| 2022 prediction | 41 | 0.9956 | 7,958,450 | 5,340,838 | 80.35 |

results validate the applicability of the proposed approach for real-world trade balance forecasting under diverse economic conditions.

## 5.8 Model suitability with respect to trade data characteristics

International trade balance data are characterized by pronounced non-linearity, high-dimensional feature interactions, temporal dependencies, and implicit relational structures among countries. These properties make accurate modeling challenging and require learning frameworks that can effectively capture complex dependencies.

Linear models such as Ridge Regression rely on linear assumptions and treat features as independent, limiting their ability to model complex economic interactions. As a result, Ridge achieved a relatively low predictive performance with an $R^2$ score of 0.421. Similarly, Support Vector Regression (SVR), although capable of modeling non-linear relationships through kernel functions, is sensitive to hyperparameter selection and feature scaling, leading to moderate performance with an $R^2$ of 0.488.

Tree-based ensemble methods, including Random Forest, Gradient Boosting, and XGBoost, are more effective at capturing non-linear feature interactions. These models achieved improved performance, with $R^2$ scores of 0.556, 0.583, and 0.601, respectively. However, these approaches treat each country as an independent sample and fail to explicitly model inter-country economic dependencies, which are fundamental to global trade systems.

Neural network-based models further enhance modeling flexibility. The Multilayer Perceptron (MLP) achieved an $R^2$ of 0.612 by learning complex non-linear feature representations. The LSTM network, designed to capture temporal dependencies, achieved a slightly higher $R^2$ of 0.629, benefiting from historical trade patterns. Nevertheless, both models rely on vectorized inputs and do not incorporate relational structures among countries.

In contrast, the proposed GNN explicitly models countries as nodes and their trade-related similarities as edges, enabling direct learning of relational, topological, and feature-level dependencies. This structural alignment with the underlying trade network allows the proposed model to achieve superior predictive performance, with an $R^2$ of 0.9158, an RMSE of 0.28, and an MAE of 0.19. The results demonstrate that while conventional and deep learning models partially address non-linearity and temporal dynamics, only the graph-based formulation fully captures the relational nature of international trade, leading to statistically and practically superior performance.

## 5.9 Model interpretability: SHAP analysis

To enhance the transparency of the proposed GNN-based trade balance forecasting model, SHAP (SHapley Additive exPlanations) analysis is employed to quantify the contribution of individual features to the model's predictions. SHAP values are computed on the test dataset using a kernel-based explainer, allowing model-agnostic interpretation of the learned relationships. In total, 25 input features are considered, and SHAP values are evaluated for 21 test samples.

**5.9.1 SHAP-based feature importance.** Table 5 presents the top 20 most influential features ranked by their mean absolute SHAP values. The results indicate that squared trade balance and export-related features play a dominant role in shaping the model's predictions. In particular, *export_2022_squared* and *balance_2022_squared* emerge as the most influential variables, highlighting the importance of non-linear transformations of recent trade statistics. Additionally, historical export values and balance ratios across multiple years significantly contribute to forecasting accuracy, demonstrating the model's ability to leverage temporal trade dynamics.

**5.9.2 Comparison with alternative importance measures.** To validate the robustness of the SHAP-based interpretation, feature importance is further examined using Random Forest and permutation-based importance methods. S1 Table shows that historical trade balance values and global export share consistently rank among the most influential features, aligning closely with the SHAP results. Similarly, permutation importance confirms the strong impact of squared balance and recent export values, reinforcing the reliability of the interpretability findings.

**5.9.3 SHAP visualization.** Fig 11 provides a global SHAP summary visualization, illustrating both the magnitude and direction of feature contributions across the test samples. Features related to recent export values, trade balance ratios, and global market share exhibit strong and consistent influence, confirming that the proposed GNN model relies on economically meaningful indicators rather than spurious correlations.

# 6 Discussion

The experimental results demonstrate that the proposed GNN-based framework provides a robust and reliable solution for trade balance prediction by effectively capturing complex, non-linear, and relational dependencies inherent in global trade systems. The model achieved the highest predictive accuracy on the 2023 test set with an

**Table 5. Top 20 features ranked by mean absolute SHAP values.**

| Feature | SHAP importance |
|---|---|
| export_2022_squared | $8.62 \times 10^6$ |
| balance_2022_squared | $7.92 \times 10^6$ |
| Value exported in 2021 (USD thousand) | $4.56 \times 10^6$ |
| Share in world exports (%) | $3.82 \times 10^6$ |
| balance_ratio_2020 | $3.50 \times 10^6$ |
| balance_ratio_2021 | $3.29 \times 10^6$ |
| balance_ratio_2019 | $3.08 \times 10^6$ |
| Value exported in 2019 (USD thousand) | $2.85 \times 10^6$ |
| Value exported in 2023 (USD thousand) | $2.53 \times 10^6$ |
| Value exported in 2020 (USD thousand) | $2.04 \times 10^6$ |
| Trade balance in 2021 (USD thousand) | $1.87 \times 10^6$ |
| Trade balance in 2019 (USD thousand) | $1.84 \times 10^6$ |
| Trade balance in 2020 (USD thousand) | $1.84 \times 10^6$ |
| Annual growth in value (2019–2023) (%) | $1.71 \times 10^6$ |
| Trade balance in 2022 (USD thousand) | $1.18 \times 10^6$ |
| balance_ratio_2022 | $1.04 \times 10^6$ |
| Value exported in 2022 (USD thousand) | $9.64 \times 10^5$ |
| balance_growth_21_22 | $6.14 \times 10^5$ |
| Concentration of importing countries | $5.03 \times 10^5$ |
| growth_acceleration | $4.03 \times 10^5$ |

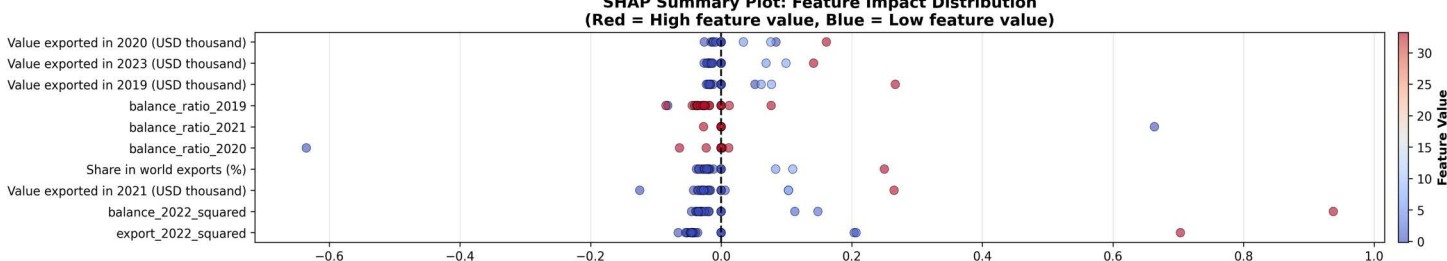

**Fig 11. SHAP summary plot showing the global feature importance and contribution patterns for trade balance prediction using the proposed GNN model.**

$R^2$ score of 0.9158, outperforming all baseline methods, while maintaining stable error margins (RMSE of 96.21 million USD and MAE of 28.94 million USD). Statistical significance analysis further supports these findings, with non-parametric tests confirming consistent performance superiority across experimental runs ($p < 0.001$), indicating that the observed improvements are not driven by stochastic variation. Sensitivity analysis highlights the model's robustness across diverse economic conditions, as no statistically significant performance difference was observed between OECD and non-OECD country groups ($p = 0.1248$), despite notable differences in economic scale and sample availability. Moreover, strong temporal generalization was observed across historical evaluations from 2019 to 2022, where the model consistently achieved high predictive accuracy ($R^2 > 0.98$), demonstrating resilience to changes in feature availability and economic volatility. Compared with traditional machine learning models such as Random Forest and XGBoost, and deep learning approaches including MLP and LSTM, which primarily operate under independence or sequential assumptions, the proposed GNN explicitly models inter-country relationships, allowing it to better reflect the interconnected nature of global trade. These findings are consistent with emerging literature that successfully applies graph-based methods to economic and trade forecasting [5,40], and suggest that graph-based learning offers a meaningful advancement for trade forecasting applications, with strong potential for real-world policy analysis and decision-support systems.

## 7  Concluding remarks

This study demonstrates the effectiveness of machine learning for modeling the complex dynamics of global trade balance prediction, where outcomes are shaped by geopolitical events, policy interventions, and volatile market conditions. By adopting a structured and carefully validated learning pipeline, the proposed GNN model achieves superior predictive performance compared to conventional machine learning and deep learning baselines, highlighting its ability to capture non-linear relationships and inter-country dependencies inherent in trade data. While ensemble methods such as Random Forests provide competitive results by modeling feature interactions efficiently, more complex architectures including DNNs and Transformer-based models exhibit markedly degraded performance. This performance degradation can be attributed to several factors: pronounced data heterogeneity across 229 countries, inconsistencies arising from missing-value imputation, and limited sample availability, all of which hinder the stable optimization of high-capacity models. Furthermore, the static nature of the modeling framework restricts the ability of sequence-based architectures to fully leverage temporal dependencies or adapt to structural breaks induced by economic shocks and policy shifts. In contrast, the GNN's relational inductive bias enables effective information sharing across economically connected entities, improving generalization and robustness. These findings underscore the importance of aligning model complexity with data characteristics and structural properties, and they support the adoption of graph-based learning as a flexible, data-driven approach for advancing trade forecasting and decision-support systems.

## 8 Acronyms

Graph Neural Network (GNN), artificial neural networks (ANNs), AutoRegressive Integrated Moving Average (ARIMA), International Trade Centre (ITC), Conference on Trade and Development (UNCTAD), World Trade Organization (WTO), Graph Attention Network (GAT), Support Vector Regression (SVR), Exponential Linear Units (ELU), multilayer perceptron (MLP), mean squared error (MSE), Root Mean Squared Error (RMSE), Deep Neural Network (DNN), SHapley Additive exPlanations (SHAP).

## Author contributions

**Conceptualization:** Yifei Huang, Cheng Ding.

**Data curation:** Yifei Huang, Zhiyuan He.

**Formal analysis:** Zhiyuan He, Cheng Ding.

**Methodology:** Yifei Huang.

**Supervision:** Cheng Ding.

**Validation:** Yifei Huang, Zhiyuan He, Cheng Ding.

**Visualization:** Yifei Huang, Zhiyuan He.

**Writing – original draft:** Yifei Huang, Zhiyuan He, Cheng Ding.

**Writing – review & editing:** Yifei Huang, Cheng Ding.

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
