## [Decision Letter · Decision Letter 0]

17 Oct 2025

PONE-D-25-41652

A Random Forest Regression-based Approach for Accurate Trade Balance Forecasting

PLOS ONE

Dear Dr. Ding,

Thank you for submitting your manuscript to PLOS ONE. After careful consideration, we have decided that your manuscript does not meet our criteria for publication and must therefore be rejected.

Specifically:

Reject recommended as the article largely failed to meet PLOS One publication criteria 3 - **Experiments, statistics, and other analyses are performed to a high technical standard and are described in sufficient detail** with reference to reviewers comments. Given the strict time frame, there is no certainty that the authors will be able to provide a robust revision to meet the above standard. More details: with reference to reviewers comments. Given the strict time frame, there is no certainty that the authors will be able to provide a robust revision to meet the above standard. More details: with reference to reviewers comments. Given the strict time frame, there is no certainty that the authors will be able to provide a robust revision to meet the above standard. More details: with reference to reviewers comments. Given the strict time frame, there is no certainty that the authors will be able to provide a robust revision to meet the above standard. More details: Criteria for Publication | PLOS One

I am sorry that we cannot be more positive on this occasion, but hope that you appreciate the reasons for this decision.

Kind regards,

Elochukwu Ukwandu, PhD

Academic Editor

PLOS ONE

Reviewers' comments:

Reviewer's Responses to Questions

**Comments to the Author**

1. Is the manuscript technically sound, and do the data support the conclusions?

Reviewer #1: Yes

Reviewer #2: Partly

2. Has the statistical analysis been performed appropriately and rigorously?

Reviewer #1: Yes

Reviewer #2: No

3. Have the authors made all data underlying the findings in their manuscript fully available?

Reviewer #1: Yes

Reviewer #2: Yes

4. Is the manuscript presented in an intelligible fashion and written in standard English?

Reviewer #1: Yes

Reviewer #2: Yes

Reviewer #1: The author presented very interesting study in the domain of "A Random Forest Regression-based Approach for Accurate Trade Balance Forecasting" so i recommend Major revision before the final approval of the manuscript.

1- ROC and AUC are classification metrics, not regression metrics. Using them for regression (Section 5.6) is methodologically incorrect and shows conceptual misunderstanding, Kindly revise and correct it

2-The model explains only 25% of variance too low to claim accurate forecasting. conclusions overstate model performance.

3-The manuscript alternates between predicting “trade balance” and “export value.” Section 4.3 defines the target as export value, contradicting the abstract and other sections.

4-Despite poor R², the study repeatedly claims Random Forest “outperforms” deep models and provides “accurate predictions.

5-There’s no significance analysis for performance differences between models. kindly revise and correct it

6- Add acronyms section before the references.

7- Motivation behind the study is missing in the manuscript.

8-Negative R² values for DNN, Transformer, and Ensemble models (Table 1) indicate poor model fitting. Such results should trigger re-evaluation, not comparison for superiority.

Decision: Major Revision

Reviewer #2: This manuscript presents a comparative study of machine learning models for forecasting global trade balances. The topic is relevant, and the scale of the study (229 countries) is a notable contribution. However, the manuscript requires significant revisions to strengthen its methodological rigor, contextualization within the existing literature, and the depth of its analysis and discussion before it can be considered for publication.

1. Lack of Methodological and Literature Context:

Methodological Rationale: The manuscript would be significantly strengthened by including a dedicated section that justifies the selection of the specific machine learning models (e.g., DNN, Transformer, ensemble, and Random Forest). The authors should explain the anticipated strengths of each model in the context of trade data characteristics (e.g., non-linearity, high dimensionality, temporal aspects).

Critique of Previous ML Applications: The introduction and literature review should add more critical synthesis of previous studies that used ML for trade forecasting. Specifically, the authors should clearly delineate the common methodological limitations in prior work (e.g., over-reliance on single models, inadequate feature engineering, lack of interpretability) to better position their own research contributions.

2. Inadequate Description of Experimental Setup:

Data Partitioning and Validation: The manuscript misses a critical description of the data splitting strategy (e.g., train/validation/test sets) and the validation protocol. It is essential to detail whether a temporal split was used to preserve the time-series nature of the data and to explicitly state the measures taken to prevent data leakage, as this is fundamental to the credibility of the results.

3. Lack of Robustness and Interpretability Analysis:

Sensitivity Analysis: To validate the robustness of the findings, the authors should perform sensitivity analyses; like:

Geographical Robustness: Re-running the analysis on economically distinct subgroups (e.g., OECD vs. non-OECD nations).

Temporal Robustness: Evaluating model performance on different time periods to test for temporal consistency.

Methodological Robustness: Repeating the feature selection process with alternative methods to ensure the stability of the selected features.

Model Interpretability: The value of the study for policymakers and economists would be greatly enhanced by incorporating model interpretation techniques. Employing SHAP (SHapley Additive exPlanations) analysis or presenting normalized variable importance plots would help answer critical questions about which economic indicators are most influential globally and the direction of their impact on trade balances.

4. Insufficient Discussion and Acknowledgment of Limitations:

Discussion Section: The manuscript is currently missing a dedicated discussion section. This section should interpret the results in the context of the research questions and the existing literature. The striking performance gap between Random Forest and all other models (point #6) requires particularly careful interpretation.

Study Limitations: A balanced academic paper must acknowledge its limitations. The authors should discuss:

* The modest explanatory power (R² = 0.254) of the best model, indicating significant unexplained variance.

* The potential reasons for the catastrophic failure of complex models like DNN and Transformer.

* Challenges associated with data heterogeneity and imputation across 229 countries.

* The limitations of a static modeling approach that may not account for temporal dependencies and structural breaks.

5. Results Presentation and Clarity:

Table 1 & 2: The results show that all models except Random Forest perform catastrophically poorly (with extreme negative R² values). The authors must explicitly address this in the text, explaining that these models failed to outperform a simple mean baseline and discussing the potential reasons for such a significant performance disparity. Presenting this as a standard model comparison is misleading.

Figure 4: The x-axis labels are currently overlapping, obstructing visual understanding. This figure should be reformatted for clarity.

Figure 5: The caption for Figure 5 should be clarified. It is currently unclear what the "country feature" represents—does it indicate results for a specific country, or a specific feature related to countries?

**Recommendation**

**Major Revision.** The manuscript has the potential to be a valuable contribution but requires substantial revisions to address the major concerns outlined above, particularly regarding methodological justification, validation, interpretability, and the discussion of results and limitations.

.

Reviewer #1: No

Reviewer #2: No

- - - - -

---

## [Author Response · Author response to Decision Letter 1]

31 Jan 2026

Thanks for the comments, we have address all of them and included the response in the attached file.

---

## [Decision Letter · Decision Letter 1]

13 Mar 2026

Dear Dr. Ding,

Thank you for submitting your manuscript to PLOS ONE. After careful consideration, we feel that it has merit but does not fully meet PLOS ONE’s publication criteria as it currently stands. Therefore, we invite you to submit a revised version of the manuscript that addresses the points raised during the review process.

We look forward to receiving your revised manuscript.

Kind regards,

John Sum, Ph.D.

Academic Editor

PLOS One

Journal Requirements:

1.Please ensure that your manuscript meets PLOS ONE's style requirements, including those for file naming. The PLOS ONE style templates can be found at https://journals.plos.org/plosone/s/file?id=wjVg/PLOSOne_formatting_sample_main_body.pdf and https://journals.plos.org/plosone/s/file?id=ba62/PLOSOne_formatting_sample_title_authors_affiliations.pdf

3. Please note that your Data Availability Statement is currently missing the repository name and/or the DOI/accession number of each dataset OR a direct link to access each database. If your manuscript is accepted for publication, you will be asked to provide these details on a very short timeline. We therefore suggest that you provide this information now, though we will not hold up the peer review process if you are unable.

4. In the online submission form, you indicated that your data is available only on request from a third party. Please note that your Data Availability Statement is currently missing the name of the third party contact or institution / contact details for the third party, such as an email address or a link to where data requests can be made. Please update your statement with the missing information.

5. We note that your manuscript is not formatted using one of PLOS ONE’s accepted file types. Please reattach your manuscript as one of the following file types: .doc, .docx, .rtf, or .tex (accompanied by a .pdf).

If your submission was prepared in LaTex, please submit your manuscript file in PDF format and attach your .tex file as “other.”

Additional Editor Comments (if provided):

As a reviewer still has some comments, the authors please revise your manuscript in accordance with the comments addressed.

Reviewers' comments:

Reviewer's Responses to Questions

**Comments to the Author**

Reviewer #1: All comments have been addressed

Reviewer #2: All comments have been addressed

2. Is the manuscript technically sound, and do the data support the conclusions?

Reviewer #1: Yes

Reviewer #2: Yes

3. Has the statistical analysis been performed appropriately and rigorously?

Reviewer #1: Yes

Reviewer #2: Yes

4. Have the authors made all data underlying the findings in their manuscript fully available?

Reviewer #1: Yes

Reviewer #2: Yes

5. Is the manuscript presented in an intelligible fashion and written in standard English?

Reviewer #1: Yes

Reviewer #2: Yes

Reviewer #1: I accept the article as authors addressed my all the comments carefully, no further comments from my side.

Reviewer #2: Thank you for your efforts in revising the manuscript. I believe the work is now very close to being suitable for publication. However, I have identified a few minor issues that should be addressed to improve the overall presentation:

Typographical Error: There is a spelling error in Figure 1. Please review the figure caption or the text within the figure for the indicated misspelling and correct it.

Figure Readability: In Figure 3, the text/levels appear to be overlapping, which makes them difficult to read. Please adjust the layout, font size, or spacing to ensure all elements are clearly legible.

Manuscript Length / Supplementary Material: The main text currently contains a high density of plots and tables. To improve the flow for the reader, I recommend moving some of the less critical plots or supplementary analyses to the Supplementary Materials section, keeping only the most essential figures in the main body.

Strengthening the Discussion: The final paragraph of the Discussion section currently makes a broad claim about the model's potential for policy analysis. To strengthen this, the authors should support their findings by citing specific references. For example, rather than stating that the findings are "consistent with emerging literature," please cite 1-2 specific papers that have similarly applied graph-based methods to economic or trade forecasting problems. This will ground your claim in the existing body of work.

.

Reviewer #1: No

Reviewer #2: No

---

## [Author Response · Author response to Decision Letter 2]

16 Mar 2026

Thanks for the comments from the reviwers. We have addressed them and the responses are in the attached file.

---

## [Editor Report · Decision Letter 2]

19 Mar 2026

A GNN-based Approach for Accurate Trade Balance Forecasting and Interpretable Analysis

PONE-D-25-41652R2

Dear Dr. Ding,

We’re pleased to inform you that your manuscript has been judged scientifically suitable for publication and will be formally accepted for publication once it meets all outstanding technical requirements.

Kind regards,

John Sum, Ph.D.

Academic Editor

PLOS One
---

## [Editor Report · Acceptance letter]

PONE-D-25-41652R2

PLOS One

Dear Dr. Ding,

I'm pleased to inform you that your manuscript has been deemed suitable for publication in PLOS One. Congratulations! Your manuscript is now being handed over to our production team.

Kind regards,

on behalf of

Prof. John Sum

Academic Editor

PLOS One